# V²A-Mark: Versatile Deep Visual-Audio Watermarking for Manipulation Localization and Copyright Protection

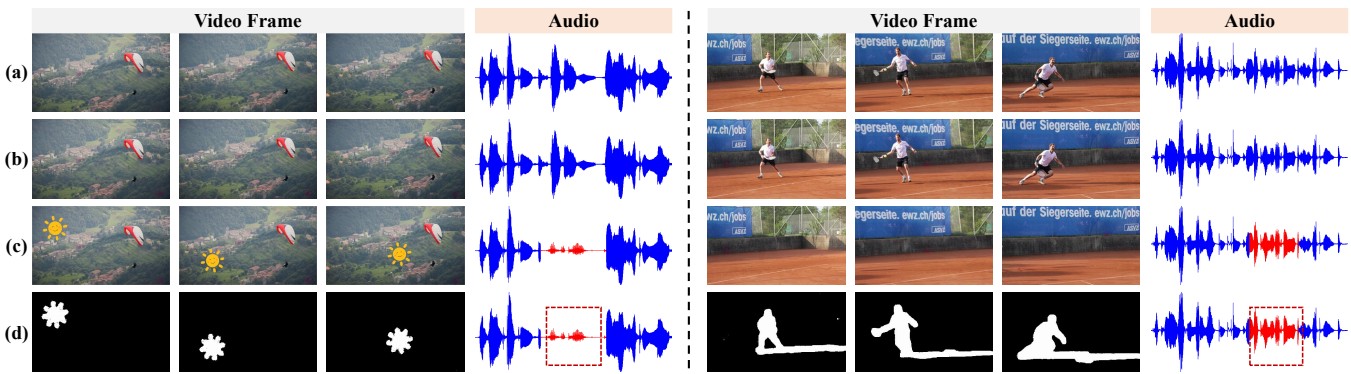

Figure 1: Two application instances of V²A-Mark. (a): Original video, (b): Watermarked video, (c) Tampered video, (d) Tampered visual areas and audio period. We propose a versatile deep visual-audio proactive forensics framework, dubbed V²A-Mark. Our method can embed an invisible cross-modal watermark into the original video frames and audio (a), producing watermarked video frames and audio (b). If they are tampered by object removal, copy-and-paste, or any editing methods during network transmission (c), we can accurately get the predicted visual tampered areas, audio tampered periods, and the copyright (d).

## ABSTRACT

AI-generated video has revolutionized short video production, film-making, and personalized media, making video local editing an essential tool. However, this progress also blurs the line between reality and fiction, posing challenges in multimedia forensics. To solve this urgent issue, V²A-Mark is proposed to address the limitations of current video tampering forensics, such as poor generalizability, singular function, and single modality focus. Combining the fragility of video-into-video steganography with deep robust watermarking, our method can embed invisible visual-audio localization watermarks and copyright watermarks into the original video frames and audio, enabling precise manipulation localization and copyright protection. We also design a temporal alignment and fusion module and degradation prompt learning to enhance the localization accuracy and decoding robustness. Meanwhile, we introduce a sample-level audio localization method and a cross-modal copyright extraction mechanism to couple the information of audio and video frames. The effectiveness of V²A-Mark has been verified on a visual-audio tampering dataset, emphasizing its superiority in localization precision and copyright accuracy, crucial for the sustainable development of video editing in the AIGC video era. [1]

## CCS CONCEPTS

• **Computing methodologies** → **Computer vision**.

## KEYWORDS

Manipulation Localization, Copyright Protection, Watermarking

## 1 INTRODUCTION

2024 is regarded as a boom year of AI-generated video. Benefited from diffusion models and the influx of extensive video data, a large amount of video generation models and editing methods [3, 7, 10, 12, 41, 55] have emerged, offering convenience in the production of short videos, film-making, advertising, and customized media. Specifically, local editing [40, 56, 59] has become a vital feature of AI video generation tools. For instance, AI dubbing software, capable of altering the facial expressions, lip movements, and voices of characters in a video, is extensively used in simultaneous interpretation and movie dubbing. However, this powerful editing capability is a double-edged sword. It not only facilitates video editors and creators but also blurs the boundaries between reality and forgery, posing new challenges for tamper forensics. Therefore, it is urgent to develop a method for visual-audio tamper localization and copyright protection, which can be widely used in court evidence, rumor verification, and beyond.

Most visual-audio manipulation localization methods [27, 28, 32, 36, 45, 54] are passive, which mainly rely on excavating the temporal and spatial anomalous traces from the suspect videos themselves to predict tampered regions. However, these methods often prove ineffective against AIGC-based video tampering, which exhibits fewer artifacts and more realistic texture details. Additionally, most passive black-box localization networks typically require the introduction of specific types of manipulation during training, rendering

---

[1]For reproducible research, the complete source code with all pre-trained model weights of our proposed V²A-Mark will be made publicly available.

them ineffective against previously unseen editing methods. Therefore, these methods have obvious shortcomings in generalization ability and accuracy of manipulation localization.

Given the inherent drawbacks of passive detection and localization, visual-audio watermarking has become a consensus technology for proactive forensics. However, existing video watermarking methods are fraught with some issues. **1) Poor Accuracy**: Although traditional fragile watermarking methods [13, 25] can achieve block-wise manipulation location via hash verification, their accuracy is unsatisfactory and difficult to reach the pixel-wise localization. **2) Singular Function**: Video manipulation localization and copyright protection tend to be treated as two distinct and separate tasks. Tampering forensic methods lack the capability for copyright protection, limiting the applicative value of their prediction results. Simultaneously, robust deep video watermarking methods [30, 58] can only provide copyright protection and are unable to precisely pinpoint the locations of tampering within videos. **3) Single Modality**: Most current forensic methods often only focus on a single visual [60] or audio modality [39] and have not established effective cross-modal interaction mechanisms. How to effectively utilize cross-modal information for manipulation localization and cross-verification of copyrights is an urgent issue.

To address the above-mentioned issues, we propose an innovative multi-functional and multi-modal watermarking method, dubbed $V^2$**A-Mark**. In the visual section, integrating the fragility of video-into-video steganography and the robustness of bit-into-video watermarking, we simultaneously embed both localization and copyright watermarks into the video frames, enabling the decoding network to independently extract tampered areas and copyright information. In the audio section, we insert a versatile watermark into the host audio and use it to assist in the reconstruction of visual copyright information, while identifying the tampered periods in the audio. Thus, our contributions are as follows.

❑ (1) We design an innovative deep versatile, cross-modal video watermarking framework, dubbed $V^2$**A-Mark**, for visual-audio manipulation localization and copyright protection. It can embed invisible localization and copyright watermarks into video frames and audio samples simultaneously, and then obtain visual tampered area, audio tampered period, and exact copyright information in the decoding end.

❑ (2) In the visual section, we develop a **temporal alignment and fusion module** (TAFM) and a **degradation prompt learning** (DPL) mechanism, enabling the network to fully leverage temporal information for high-fidelity concealment and robust prediction of localization and copyright results.

❑ (3) In the audio section, we embed sample-level versatile watermarks into the pristine audio to identify the tampered samples and extract the copyright information. Furthermore, a cross-modal extraction mechanism is proposed to obtain the final copyright from the information of audio and video frames.

❑ (4) The effectiveness of our method has been verified on our constructed visual-audio tampering dataset. Compared to other approaches, our method has notable merits in localization accuracy, generalization abilities, and copyright precision without any labeled data or additional training required for specific tampering types.

## 2 RELATED WORKS

### 2.1 Manipulation Localization

Prevalent image forensic techniques have focused on localizing specific types of manipulations via exploring artifacts and anomalies in tampered images [6, 14, 20, 22, 23, 28, 42, 46, 48, 52–54]. Recently, HiFi-Net [11] used multi-branch feature extractor and localization modules for AIGC-synthesized and edited images. SAFL-Net [42] designed a feature extraction approach to learn semantic-agnostic features with specific modules and auxiliary tasks. IML-ViT [32] firstly introduced vision transformer for image manipulation localization and modified ViT components to address three unique challenges in high resolution, multi-scale, and edge supervision. MaLP [2] introduced a large number of forgery images to learn the matched template and localization network. Targeted at video tamper localization, [45] exploited the spatial and temporal traces left by inpainting and guided the extraction of inter-frame residual with optical-flow-based frame alignment. UVL [36] designed a novel hybrid multi-stage architecture that combines CNNs and ViTs to effectively capture both local and global video features. However, the above-mentioned passive localization methods are often limited in terms of generalization and localization accuracy, which usually work on known tampering types that have been trained.

### 2.2 Video Watermarking and Steganography

Video watermarking is a widely accepted forensic method, which can be broadly utilized for the verification, authenticity, and traceability of images. In terms of robustness levels for extraction, video watermarking can be divided into fragile and robust watermarking [21, 51, 60]. Although classical fragile watermarking [13, 15, 18, 25, 26, 34, 35, 43] can achieve block-wise tamper localization, their localization accuracy and flexibility are unsatisfactory. Therefore, how to realize joint pixel-level tamper localization and copyright protection has still a lot of room for research.

Thanks to the development of deep learning, learning-based video watermarking has attracted increased attention. For deep robust video watermarking, an intuitive approach is to apply image watermarking methods [16, 31, 61] frame by frame. For instance, HiDDeN [61] firstly designed a deep encoder-decoder network to hide and recover bitstream. Moreover, many differentiable distortion layers such as JPEG compression, screen-shooting, and face swapping [1, 8, 29, 47] were incorporated to enhance the robustness of the encoder-decoder watermarking framework. Meanwhile, CIN [31] and FIN [9] utilized flow-based models to further improve the fidelity of container images. However, these deep watermarking methods have a singular function and cannot accurately localize the tampered areas. Moreover, there are other explorations to address video degradation and temporal correlations. For instance, DVMark [30] used an end-to-end trainable multi-scale network for robust watermark embedding and extraction across various spatial-temporal clues. REVMark [58] focused on improving the robustness against H.264/AVC compression via the temporal alignment module and DiffH264 distortion layer. LF-VSN [33] utilized invertible blocks and the redundancy prediction module to realize large-capacity and flexible video steganography.

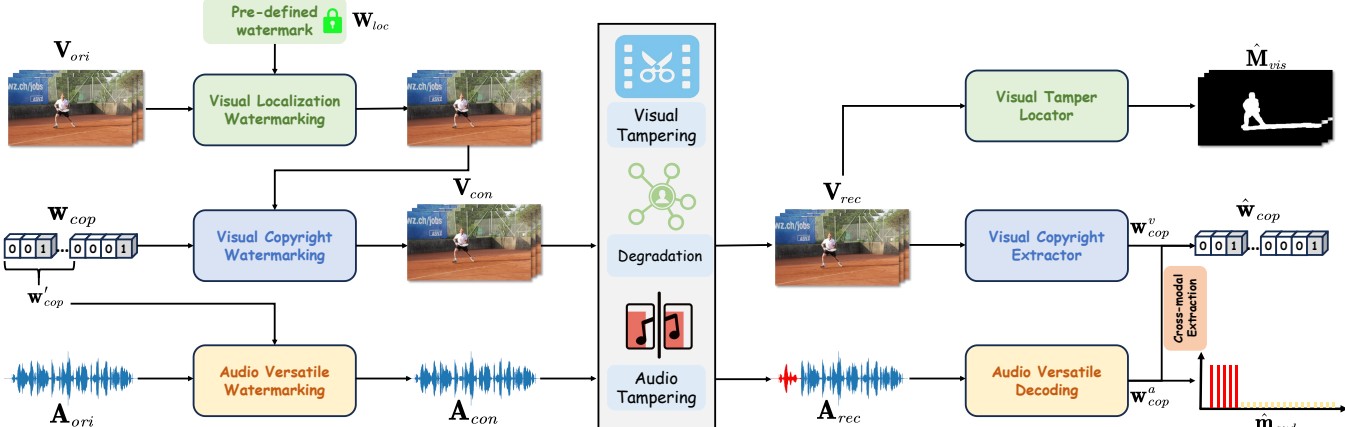

**Figure 2: Overall Framework of our proposed V²A-Mark. We embed pre-defined visual localization watermark $W_{loc}$, copyright watermark $w_{cop}$ and audio versatile watermark $w'_{cop}$ into the original video frames and audio to produce $V_{con}$ and $A_{con}$. If undergoing malicious tampering, we can still extract exact copyright $\hat{w}_{cop}$, visual tampered masks $\hat{M}_{vis}$ and audio tampered periods $\hat{m}_{aud}$. Note that $\hat{w}_{cop}$ is obtained via our cross-modal extraction mechanism, combining $w^a_{cop}$ and $w^v_{cop}$.**

## 3 METHODS

### 3.1 Overall Framework of V²A-Mark

To achieve multimodal, versatile, and proactive manipulation localization and copyright protection, as shown in Fig. 2, the proposed V²A-Mark consists of two key sections, namely the visual hiding and decoding (Sec. 3.3), and the audio hiding and decoding (Sec. 3.4). In the visual hiding section, we sequentially embed pre-defined visual localization watermarks $W_{loc} \in \mathbb{R}^{H \times W \times T \times C}$ and the copyright watermark $w_{cop} \in \{0, 1\}^k$ into the original video sequences $V_{ori} \in \mathbb{R}^{H \times W \times T \times C}$ to get the container video $V_{con} \in \mathbb{R}^{H \times W \times T \times C}$. In the audio hiding section, we add versatile watermark $w'_{cop} \in \{0, 1\}^n$ to the original audio $A_{ori} \in \mathbb{R}^L$ in a sample-level manner to obtain the $A_{con} \in \mathbb{R}^L$. Note that $T$ and $L$ denote the number of video frames and length of the audio, respectively. "Versatile" means that this audio watermarking and decoding module can achieve audio manipulation localization and copyright protection at the same time. Moreover, the potential impacts on container videos during network transmission can be divided into two types, namely malicious tampering and common degradation. Thus, the network transmission pipeline of video frames and audio is modeled as follows.

$$V_{rec} = \mathcal{D}_v(V_{con} \odot (1 - M) + \mathcal{T}_v(V_{con}) \odot M), \quad (1)$$

$$A_{rec} = \mathcal{D}_a(A_{con} \odot (1 - m) + \mathcal{T}_a(A_{con}) \odot m), \quad (2)$$

where $\mathcal{T}_v(\cdot)$ and $\mathcal{T}_a(\cdot)$ respectively denote the video and audio manipulation function. $\mathcal{D}_v(\cdot)$ and $\mathcal{D}_a(\cdot)$ respectively denote the video and audio degradation operation. $M \in \mathbb{R}^{H \times W \times T}$ and $m \in \mathbb{R}^L$ respectively denote the tempered visual masks and audio periods.

Upon receiving the video $V_{rec}$ and audio $A_{rec}$, we attempt to recover the previously embedded watermarks on different robustness levels and conduct corresponding forensics based on the extracted watermarks. In the visual decoding section, our framework precisely extracts the tampered video masks $\hat{M}_{vis}$ and copyright information $w^v_{cop}$. Concurrently, as shown in Fig. 2, the tampered periods $\hat{m}_{aud}$ and the copyright $w^a_{cop}$ in the audio will be extracted from the audio

versatile decoding module. The final restored copyright information of the video $\hat{w}_{cop}$ will be obtained via cross-modal combination of $w^a_{cop}$ and $w^v_{cop}$ (Sec. 3.5). Finally, the visual-audio tamper forensics process of V²A-Mark can be categorized into several scenarios, where $\wedge$ and $\vee$ respectively denote the "element-wise and" and "element-wise or".

❏ (1) $\hat{w}_{cop} \not\approx w_{cop}$: Suspicious $V_{rec}$ was not processed via our V²A-Mark, and we are also unable to ascertain the authenticity of the corresponding audio $A_{rec}$. They cannot be used as evidence.

❏ (2) $\hat{w}_{cop} \approx w_{cop} \wedge (\hat{M}_{vis} \not\approx 0 \vee \hat{m}_{aud} \not\approx 0)$: Suspicious $V_{rec}$ or $A_{rec}$ has undergone tampering, disqualifying it as valid evidence.

❏ (3) $\hat{w}_{cop} \approx w_{cop} \wedge \hat{M}_{vis} \approx 0 \wedge \hat{m}_{aud} \approx 0$: Suspicious $V_{rec}$ and $A_{rec}$ are both credible and have not been tampered with. V²A-Mark ensures the authenticity and integrity of this video.

### 3.2 Preliminaries and Motivations

Previous work EditGuard [57] has already validated the feasibility of using the fragility and locality of image-into-image steganography for proactive image tamper localization. Specifically, **fragility** means the damage to the container image results in corresponding damage to the revealed secret image. **Locality** indicates that damage to the container image and the revealed secret image is essentially pixel-level and directly correlated. These two properties can also be effectively applied in proactive video localization. Meanwhile, EditGuard [57] adopts a "sequential embedding and parallel decoding" structure to realize united tamper localization and copyright protection. Clearly, one direct approach is to watermark each video frame via EditGuard. However, this method overlooks the exploitation of temporal correlation, making it challenging to ensure the robustness of the reconstructed watermarks and the temporal consistency of the watermarked videos. Therefore, the key issues addressed in this paper are: **1)** How to utilize the auxiliary information from supporting frames for watermark embedding and decoding in reference frames; **2)** How to improve the

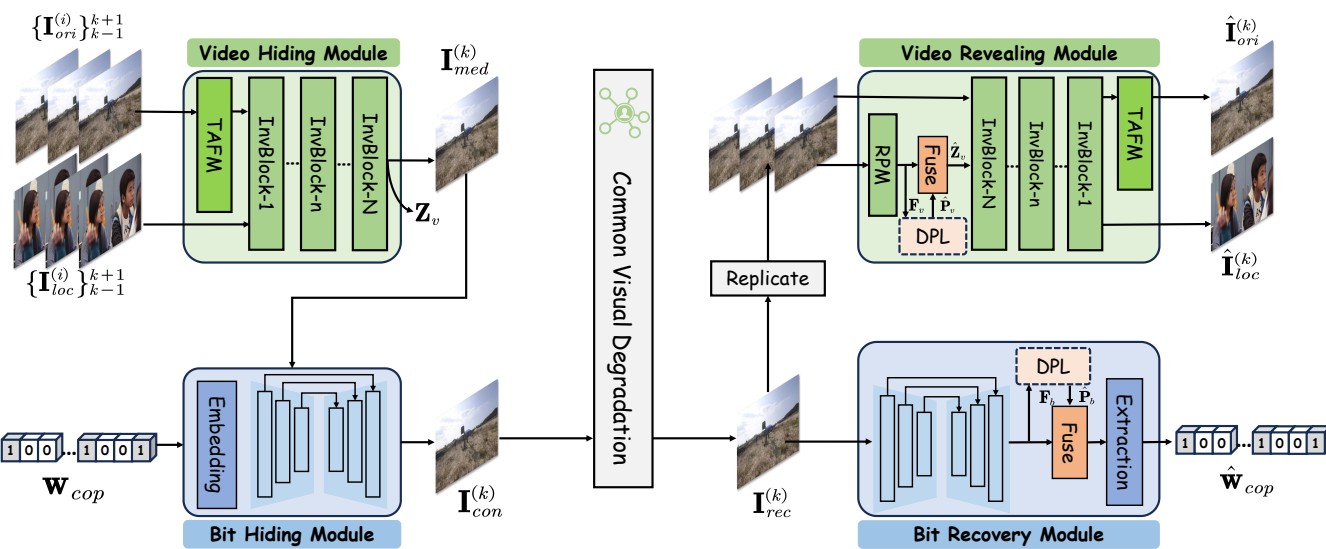

**Figure 3: Details of the network structure and training process of the proposed V²A-Mark. We design the temporal alignment and fusion module (TAFM) and degradation prompt learning (DPL) to enhance the robustness and fidelity of our method.**

robustness of existing frameworks to video degradation; 3) How to employ the watermarks embedded in video frames for audio tamper localization and copyright protection. To address the above issues, we design the visual hiding module (VHM), visual revealing module (VRM), bit hiding module (BHM), and bit recovery module (BRM). Meanwhile, we design an efficient cross-modal extraction mechanism and introduce the advanced audio versatile watermarking and decoding method [39] to achieve cross-modal tamper localization and copyright protection.

## 3.3 Visual Hiding and Decoding

### 3.3.1 Input and Output Design of Visual Section.
To achieve memory-efficient hiding and decoding, our V²A-Mark employs a multi-frame input, single-frame output structure. As shown in Fig. 3, the visual hiding is operated group-by-group via a sliding window, traversing each video frame from head to tail. We set the length of a sliding window to 3. Given the original video group $\{\mathbf{I}_{ori}^{(i)}\}_{k-1}^{k+1}$ and localization watermark group $\{\mathbf{I}_{loc}^{(i)}\}_{k-1}^{k+1}$, we firstly use the TAFM to preprocess $\{\mathbf{I}_{ori}^{(i)}\}_{k-1}^{k+1}$ and adopt $N$ invertible blocks to generate $\mathbf{I}_{med}^{(k)}$ and its by-product $\mathbf{Z}_v$. The copyright watermark $\mathbf{w}_{cop}$ is then embedded into $\mathbf{I}_{med}^{(k)}$ via a U-Net [47], producing the final container frame $\mathbf{I}_{con}^{(k)}$. For all video frames, we embed the same copyright watermark. After network transmission, V²A-Mark decodes each received video frame $\mathbf{I}_{rec}^{(k)}$ individually. On one hand, $\hat{\mathbf{w}}_{cop}$ is extracted from $\mathbf{I}_{rec}^{(k)}$ via a U-Net and an MLP extractor. On the other hand, we replicate $\mathbf{I}_{rec}^{(k)}$ threefold and feed it into the residual prediction module (RPM) [33] to produce the missing component $\hat{\mathbf{Z}}_v$. Then, $N$ invertible blocks and the TAFM are used to reconstruct the video groups and only select the intermediate frames as the result, namely $\hat{\mathbf{I}}_{ori}^{(k)}$ and $\hat{\mathbf{I}}_{loc}^{(k)}$. Note that we introduce learned degradation prompts $\mathbf{P}_v$, $\mathbf{P}_b$ in video revealing and bit recovery modules and

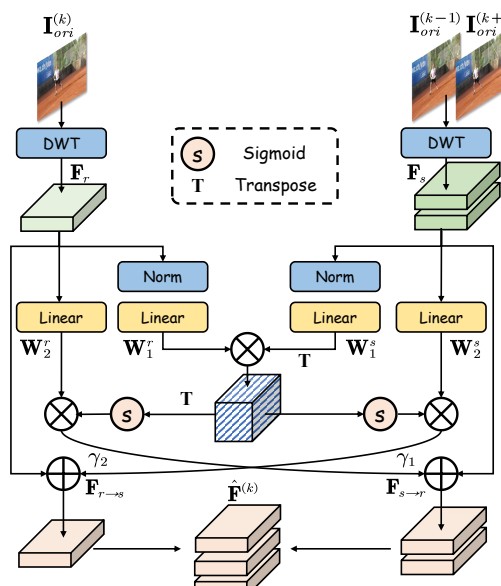

**Figure 4: Details of the proposed temporal alignment and fusion module (TAFM). It aligns the supporting frames $\mathbf{I}_{ori}^{(k-1)}$, $\mathbf{I}_{ori}^{(k+1)}$ to the reference frame $\mathbf{I}_{ori}^{(k)}$.**

fuse them with intrinsic features to further enhance the robustness of our method against common video and audio degradations.

### 3.3.2 Temporal Alignment and Fusion Module.
To further enhance the temporal consistency of the container videos, we design a temporal alignment and fusion module (TAFM) to align the supporting frames $\{\mathbf{I}_{ori}^{(i)}\}_{i\neq k}$ to the reference frame $\mathbf{I}_{ori}^{(k)}$. As shown in Fig. 4, we resort to bidirectional cross-attention mechanisms between the supporting frames and the reference frame. Specifically,

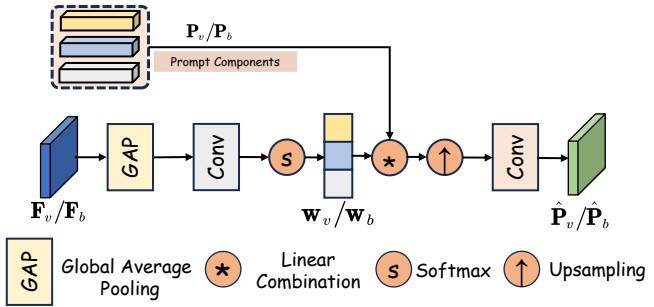

**Figure 5: Details of the proposed degradation prompt learning mechanism. It fuses the intrinsic image features $\mathbf{F}_v/\mathbf{F}_b$ with the learnable prompt components $\mathbf{P}_v/\mathbf{P}_b$ adaptively.**

we define the scaled dot production operation as follows.

$$\text{Attention}(\mathbf{Q},\mathbf{K},\mathbf{V}) = \text{softmax}\left(\mathbf{Q}\mathbf{K}^T/\sqrt{D}\right)\mathbf{V}, \quad (3)$$

where $\mathbf{Q}\in\mathbb{R}^{H\times W\times D}$ is the query matrix projected by the reference frame $\mathbf{I}_{ori}^{(k)}$, and $\mathbf{K},\mathbf{V}\in\mathbb{R}^{H\times W\times D}$ are the key and value matrix produced from the supporting frames $\{\mathbf{I}_{ori}^{(i)}\}_{i\neq k}$. Given the reference feature $\mathbf{F}_r$ and the supporting feature $\mathbf{F}_s$, they are firstly layer normalized into $\overline{\mathbf{F}}_r=\text{Norm}(\mathbf{F}_r)$ and $\overline{\mathbf{F}}_s=\text{Norm}(\mathbf{F}_s)$. Then, we use linear layers to project $\overline{\mathbf{F}}_r$, $\overline{\mathbf{F}}_s$ into $D$-dimension embedding space and calculate the cross-attention maps between reference and supporting frames as follows.

$$\mathbf{F}_{r\to s} = \text{Attention}\left(\mathbf{W}_1^r\overline{\mathbf{F}}_r, \mathbf{W}_1^s\overline{\mathbf{F}}_s, \mathbf{W}_2^s\mathbf{F}_s\right), \quad (4)$$

$$\mathbf{F}_{s\to r} = \text{Attention}\left(\mathbf{W}_1^s\overline{\mathbf{F}}_s, \mathbf{W}_1^r\overline{\mathbf{F}}_r, \mathbf{W}_2^r\mathbf{F}_r\right), \quad (5)$$

where $\mathbf{W}_1^r$, $\mathbf{W}_2^r$, $\mathbf{W}_1^s$ and $\mathbf{W}_2^s$ respectively denote the projection matrices. Finally, we perform temporal fusion between the reference frame and supporting frames via the residual connection and concatenation operation.

$$\hat{\mathbf{F}}^{(k)} = \text{Concat}(\gamma_1\mathbf{F}_{s\to r} + \mathbf{F}_r, \gamma_2\mathbf{F}_{r\to s} + \mathbf{F}_s), \quad (6)$$

where $\gamma_1$ and $\gamma_2$ respectively denote the learnable parameters. With our TAFM, V²A-Mark can better exploit temporal correlations, thus achieving more effective concealment and more robust decoding.

*3.3.3 Degradation Prompt Learning.* To further improve the robustness of V²A-Mark in decoding both visual localization and copyright watermarks, we embed learnable degradation prompts $\mathbf{P}_v\in\mathbb{R}^{h_1\times w_1\times c_1\times e_1}$, $\mathbf{P}_b\in\mathbb{R}^{h_2\times w_2\times c_2\times e_2}$ into features of the bit recovery and video revealing modules, where $c_1$, $c_2$ respectively denote the channels of prompt, $e_1$, $e_2$ respectively denote the number of degradation prompt. The degradation prompt pool comprises a series of learnable embeddings, with each corresponding to a type of potential degradation. Supposing that $\mathbf{F}_v$ and $\mathbf{F}_b$ are the outputs of the RPM in the video revealing module and the U-Net in the bit recovery module in Fig. 3 respectively, we utilize a channel attention mechanism (as shown in Fig. 5) to better encourage the interaction between the input features $\mathbf{F}_v/\mathbf{F}_b$ and the degradation prompt $\mathbf{P}_v/\mathbf{P}_b$. Specifically, the features $\mathbf{F}_v/\mathbf{F}_b$ are passed to a global average pooling (GAP) layer, a 1×1 convolution, and a softmax operator to produce a set of dynamic weight coefficients $\mathbf{w}_v/\mathbf{w}_b$,

which is inspired by [38]. These coefficients are used to fuse each degradation prompt, resulting in degradation-enhanced features. Then, we utilize convolution and concatenation operations to fuse the degradation-enhanced features with the features extracted from RPM or the U-Net in BRM. Note that we learned two distinct sets of degradation prompts for visual and bit decoding, since we aim for the BRM to be absolutely robust against degradation, while the VRM should retain some fragility against tampering.

### 3.4 Audio Hiding and Decoding

Considering that video tampering is often accompanied by corresponding changes in the audio, we try to simultaneously identify the tampered areas of the audio, and utilize the extracted audio copyright to cross-verify the copyright in the video frame. To ensure the correspondence between video and audio, we set the audio versatile copyright watermark $\mathbf{w}'_{cop}$ as part of the copyright $\mathbf{w}_{cop}$ in the video frames. For instance, $\mathbf{w}_{cop}$ is a 32-bit watermark, and $\mathbf{w}'_{cop}$ is the first 16 bits of $\mathbf{w}_{cop}$. Inspired by the advanced proactive tamper localization tool Audioseal [39], we introduce an audio watermark generator and detector to achieve audio versatile watermarking and decoding shown in Fig. 2. Specifically, we utilize the watermark generator to predict an additive watermark waveform from the audio input $\mathbf{A}_{ori}$, and use a detector to output the probability $\hat{\mathbf{m}}_{aud}$ of the presence of a watermark at each sample of the container audio $\mathbf{A}_{con}$. The detector is trained with mask augmentation strategy to ensure its accuracy and robustness. Meanwhile, we add a message embedding layer [39] in the middle of the watermark generator to embed $\mathbf{w}'_{cop}$ into $\mathbf{A}_{ori}$. In the decoding end, the detector will robustly decrypt $\mathbf{w}^a_{cop}$, which will be used to combine with $\mathbf{w}^v_{cop}$ to get the final copyright $\hat{\mathbf{w}}_{cop}$.

### 3.5 Training and Inference Details

**Training:** The training process of the visual section of the proposed V²A-Mark can be divided into two steps. Given an arbitrary original image $\mathbf{I}_{med}^{(k)}$ and watermark $\mathbf{w}_{cop}$, we first train the bit hiding and recovery module via the $\ell_2$ loss.

$$\ell_{cop} = \|\mathbf{I}_{con}^{(k)} - \mathbf{I}_{med}^{(k)}\|_2^2 + \lambda\|\hat{\mathbf{w}}_{cop} - \mathbf{w}_{cop}\|_2^2, \quad (7)$$

where $\lambda$ is set to 10. Furthermore, we freeze the weights of BHM and BRM and jointly train the VHM and VRM. Given a video group $\{\mathbf{I}_{ori}^{(i)}\}_{k-1}^{k+1}$, localization watermark group $\{\mathbf{I}_{loc}^{(i)}\}_{k-1}^{k+1}$ and copyright watermark $\mathbf{w}_{cop}$, the loss is:

$$\ell_{loc} = \|\hat{\mathbf{I}}_{ori}^{(k)} - \mathbf{I}_{ori}^{(k)}\|_1 + \alpha\|\mathbf{I}_{con}^{(k)} - \mathbf{I}_{ori}^{(k)}\|_2^2 + \beta\|\hat{\mathbf{I}}_{loc}^{(k)} - \mathbf{I}_{loc}^{(k)}\|_1, \quad (8)$$

where $\alpha$ and $\beta$ are respectively set to 100 and 1. In the audio section, we use a pre-trained audio watermarking tool [39] to realize audio hiding and decoding.

**Inference:** As shown in Fig. 2 and 3, we can conduct forensics via the pre-trained components. To extract tampered masks, we compare the pre-defined watermark $\mathbf{W}_{loc}$ with the decoded one $\hat{\mathbf{W}}_{loc}$ to obtain a binary mask $\hat{\mathbf{M}}_{vis}\in\mathbb{R}^{H\times W\times T}$:

$$\hat{\mathbf{M}}_{vis}[i, j, t] = \theta_\tau(\max(|\hat{\mathbf{W}}_{loc}[i, j, t, :] - \mathbf{W}_{loc}[i, j, t, :]|)), \quad (9)$$

where $i \in [0, H)$, $j \in [0, W)$ and $t \in [0, T)$. $\theta_\tau(z) = 1 (z \geq \tau)$. $\tau$ is set to 0.2. $|\cdot|$ is an absolute value operation. The audio tampered period $\hat{\mathbf{m}}_{aud}$ is directly extracted via the audio versatile decoder. To extract

| Method | ProPainter [59] | | | | E²FGVI [24] | | | | Video Slicing | | | |
|---|---|---|---|---|---|---|---|---|---|---|---|---|
| | F1-Score | AUC | IoU | BA(%) | F1-Score | AUC | IoU | BA(%) | F1-Score | AUC | IoU | BA(%) |
| OSN [46] | 0.164 | 0.404 | 0.125 | - | 0.170 | 0.410 | 0.126 | - | 0.382 | 0.830 | 0.262 | - |
| PSCC-Net [27] | 0.275 | 0.757 | 0.186 | - | 0.273 | 0.742 | 0.174 | - | 0.559 | 0.876 | 0.419 | - |
| HiFi-Net [11] | 0.517 | 0.699 | 0.123 | - | 0.573 | 0.763 | 0.198 | - | 0.668 | 0.906 | 0.347 | - |
| IML-ViT [32] | 0.174 | 0.521 | 0.112 | - | 0.162 | 0.516 | 0.107 | - | 0.137 | 0.509 | 0.098 | - |
| EditGuard [57] | 0.924 | 0.950 | 0.866 | 99.41 | 0.923 | 0.950 | 0.865 | 99.43 | 0.922 | 0.949 | 0.861 | 99.73 |
| V²A-Mark (Ours) | **0.944** | **0.990** | **0.897** | **99.73** | **0.943** | **0.981** | **0.895** | **99.61** | **0.941** | **0.972** | **0.891** | **99.76** |

**Table 1: Comparison with other competitive tamper forensics methods under different AIGC-based video editing methods, such as ProPainter, E²FGVI, and naive slicing. Clearly, our method achieves the best localization and copyright restoration accuracy.**

precise visual copyright, we conduct bitwise voting on the copyright extracted from each frame and select the most frequently occurring 0 or 1 as the final result $\mathbf{w}_{cop}^v$. Meanwhile, we extract audio copyright $\mathbf{w}_{cop}^a \in \{0,1\}^n$ and use it to cross-verify with $\mathbf{w}_{cop}^v \in \{0,1\}^k$, getting the final result $\hat{\mathbf{w}}_{cop} \in \{0,1\}^k$. Considering that the audio copyright watermark can often be extracted more robustly and is not easily destroyed, we directly use it as the first $n$ bits in the final multimedia copyright $\hat{\mathbf{w}}_{cop}$, which typically represents the ownership of the entire multimedia asset. The remaining $k - n$ bits are taken from the extracted visual watermark $\mathbf{w}_{cop}^v$, which will be related to the information of video frames such as resolution, time length, and frame rate. The **cross-modal extraction process** is:

$$\hat{\mathbf{w}}_{cop} = \text{Concat}(\mathbf{w}_{cop}^a, \mathbf{w}_{cop}^v[n:]). \quad (10)$$

## 4 EXPERIMENTS

### 4.1 Implementation Details

We trained our V²A-Mark in the Vimeo-90K [50] **without any tampered data**. We test our method on 30 testing videos of Davis dataset [37]. All video frames have a resolution of 448×256 and consist of 50 to 100 frames. To synthesize audio, we manually extract the video captions and use them as prompts with the VALLE-E-X audio synthesis tool [44]. The Adam [19] is used for training 250$K$ iterations with $\beta_1$=0.9 and $\beta_2$=0.5. The learning rate is initialized to $1\times10^{-4}$ and decreases by half for every 30$K$ iterations, with the batch size set to 8. $N$ in Video hiding and revealing module is set to 16. The shape of two degradation prompts $\mathbf{P}_v$ and $\mathbf{P}_b$ are 36×36×72×2 and 36×36×16×6. We embed 32-bit copyright watermarks $\mathbf{w}_{cop}$ and pure blue visual localization watermarks $\mathbf{W}_{loc}$ to original videos. We use replication padding to process the first and last frame of the original video. Meanwhile, we also embed a versatile watermark $\mathbf{w}'_{cop}$ into the original audio.

### 4.2 Comparison with Visual Tamper Localization Methods

To evaluate the visual localization and copyright recovery accuracy, we compared our method with some SOTA passive methods OSN [46], PSCC-Net [27], HiFi-Net [11], IML-ViT [32] and a proactive forensics method EditGuard [57]. Despite previous research on video tamper localization [36], we can not find methods with publicly available code. Therefore, our comparative methods primarily rely on image-based tamper localization methods on a frame-by-frame prediction. For visual tamper localization, F1-score, AUC, and IoU are used to evaluate localization accuracy. For copyright

| Method | Message | PSNR (dB) | SSIM | NIQE (↓) |
|---|---|---|---|---|
| MBRS [16] | 30 bits | 26.57 | 0.908 | 6.473 |
| CIN [31] | 30 bits | **42.41** | 0.983 | 5.858 |
| PIMoG [8] | 30 bits | 37.71 | 0.971 | 8.129 |
| SepMark [47] | 30 bits | 34.86 | 0.914 | 5.321 |
| HiNet [17] | an image | 36.46 | 0.940 | 6.271 |
| LF-VSN [33] | an image | 39.93 | 0.967 | 3.827 |
| EditGuard [57] | an image | 38.53 | 0.977 | 4.919 |
| V²A-Mark | an image | 40.83 | **0.983** | **3.484** |

**Table 2: The comparisons with other watermarking methods on the visual quality of the container video $\mathbf{V}_{con}$.**

protection, bit accuracy (BA) is used to assess the copyright recovery performance. We use two SOTA deep video inpainting methods, ProPainter [59] and E²FGVI [24], and a naive slicing approach to simulate malicious tampering.

As reported in Tab. 1, our V²A-Mark achieves impressive localization performance with an F1-Score of approximately 0.95, an AUC of 0.99, and an IoU close to 0.9. In contrast, other passive localization methods, which rely solely on tampered video clues, perform poorly in localizing unseen types of manipulation. Furthermore, when using EditGuard to watermark each video frame, although it achieves satisfactory localization results, it falls short in effectively utilizing temporal information. Consequently, the IoU of the predicted masks in various tampering methods is generally about 0.03 lower than that achieved by our V²A-Mark. Additionally, our V²A-Mark achieves an over 99.5% bit accuracy across various tampering methods, which is also marginally higher than that of EditGuard. Furthermore, as shown in Fig. 6, our method has very obvious advantages over SOTA passive localization method PSCC-Net [27], which can be attributed to our proactive tamper localization mechanism. Meanwhile, since we adopted a more effective temporal alignment and fusion method, we found that in some scenes where EditGuard can only locate the rough outline of the tampering area, our V²A-Mark can still clearly predict the tampered traces.

### 4.3 Comparison with Watermarking Methods

To evaluate the visual quality of $\mathbf{V}_{con}$, we compared our V²A-Mark with other watermarking methods on 30 testing videos from DAVIS [37]. For a fair comparison, we also retrained our EditGuard on 448×256 original videos and 32 bits. Our comparison methods include the SOTA bit-hiding watermarking method [8, 16, 31, 47], large-capacity steganography method [17, 33], and a versatile image

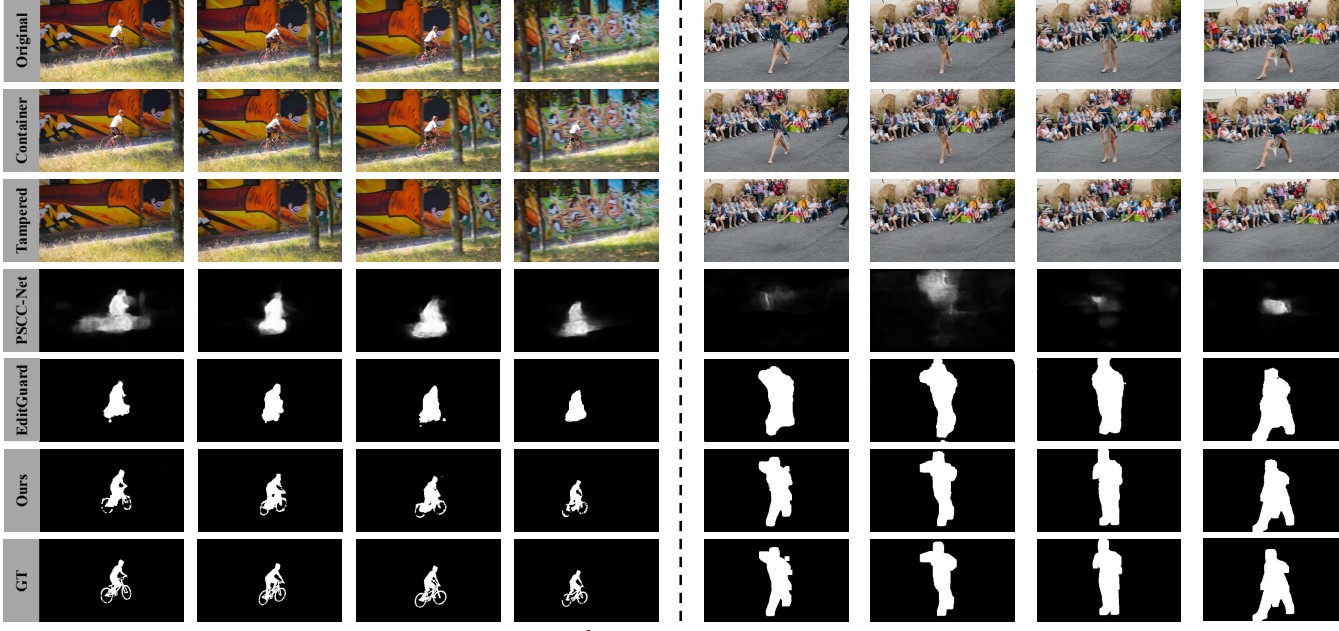

**Figure 6: Localization accuracy comparison with our V²A-Mark and other localization methods PSCC-Net [27], EditGuard [57]. Our method can predict more accurate and clearer tampered masks. We also present our container and tampered videos.**

watermarking method [57]. As shown in Tab. 2, the PSNR and SSIM of our container videos far outperform most watermarking methods such as MBRS, PIMoG, and SepMark, but is close or slightly inferior to CIN. Note that these methods only hide 30 bits in the videos, but our V²A-Mark hides both an RGB image and 32 bits. Compared with high-capacity steganography methods EditGuard, LF-VSN, and HiNet, our method also has clear advantages in visual quality. Meanwhile, the perceptual quality (NIQE) of our watermarked videos surpasses all other methods. To verify the security of our method, we perform anti-steganography detection via StegExpose [4] on container videos of various steganography methods. All the methods concealed pure blue videos into the original videos. Note that the detection set is built by mixing container and original video frames with equal proportions. We vary the detection thresholds in a wide range in StegExpose [4] and draw the ROC curve in Fig. 7. The ideal case represents that the detector has a 50% probability of detecting container videos from an equally mixed detection test, the same as a random guess. Evidently, the security of our method exhibits a significant advantage compared to all competitive methods.

## 4.4 Audio Tamper Localization

To evaluate the accuracy of V²A-Mark for audio tamper localization, we randomly insert 1s - 2s tampered audio into our constructed 30 original audio. SNR and PESQ are used to evaluate the quantitative and perceptual quality of watermarked audio. Bit accuracy is used to evaluate the bit error rate of the pre-defined $\mathbf{w}'_{cop}$ and extracted $\mathbf{w}^a_{cop}$. AUC is used to calculate the localization accuracy between the predicted audio tampered period $\mathbf{m}_{aud}$ and the ground truth of the tampered area $\mathbf{m}$. We observed from Tab. 3 that the watermarked audio maintains high SNR/PESQ on 28.29 dB/4.50 with the original audio, indicating that our V²A-Mark has little impact

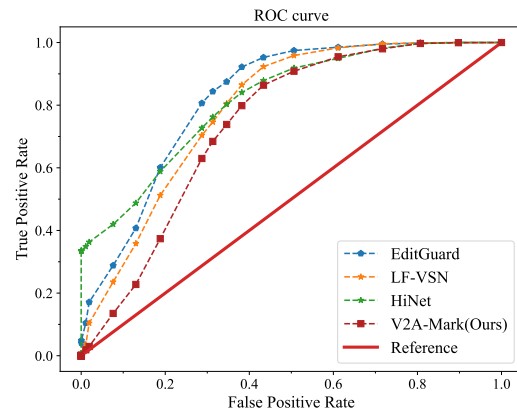

**Figure 7: ROC curve of different methods under StegExpose. The closer the curve is to the reference central axis (which means random guess), the method is better in security.**

on the semantic fidelity of the audio. Meanwhile, our method can accurately localize the tampered areas with 99.63% AUC and obtain 100% bit accuracy under "Clean" degradation, which shows that our audio localization watermark is sensitive enough to malicious tampering. Furthermore, we adopt two classical degradations on the container audio $\mathbf{A}_{con}$, namely Resample and Lowpass. "Resample" denotes resampling the container audio at a 90% sampling rate (16000Hz→14400Hz). "Lowpass" means applying low-pass filter to container audio $\mathbf{A}_{con}$, cutting frequencies above a cutoff frequency (1000Hz). As plotted in Tab. 3, although the container audio $\mathbf{A}_{con}$ has undergone different degradations, our V²A-Mark still maintains over 98% localization accuracy and nearly 100% bit accuracy, proving its robustness against common audio corruptions.

| Degradation | SNR (dB) | PESQ ($\uparrow$) | Bit. Acc. | AUC |
|---|---|---|---|---|
| Clean | 28.29 | 4.50 | 100% | 99.63% |
| Resample | - | - | 100% | 98.58% |
| Lowpass | - | - | 99.72% | 99.41% |

Table 3: Watermarked audio quality and audio tamper localization performance of our $V^2$A-Mark under clean and different degradation scenes.

| Methods | Metrics | Clean | Gaussian Noise | | H.264 | | Poisson |
|---|---|---|---|---|---|---|---|
| | | | $\sigma$=5 | $\sigma$=10 | QP=5 | QP=10 | |
| EditGuard | F1 | 0.924 | 0.891 | 0.872 | 0.900 | 0.881 | 0.896 |
| | AUC | 0.950 | 0.945 | 0.922 | 0.946 | 0.941 | 0.947 |
| | IoU | 0.866 | 0.835 | 0.812 | 0.830 | 0.828 | 0.842 |
| | BA(%) | 99.41 | 99.01 | 96.90 | 95.16 | 92.23 | 99.31 |
| $V^2$A-Mark | F1 | 0.944 | 0.904 | 0.900 | 0.915 | 0.909 | 0.913 |
| | AUC | 0.990 | 0.979 | 0.963 | 0.978 | 0.967 | 0.980 |
| | IoU | 0.897 | 0.842 | 0.833 | 0.858 | 0.850 | 0.856 |
| | BA(%) | 99.73 | 99.35 | 98.51 | 99.34 | 99.18 | 99.71 |

Table 4: Localization and bit recovery performance of our $V^2$A-Mark and EditGuard under different degradations.

## 4.5 Robustness Analysis

To analyze the robustness of our $V^2$A-Mark, we compare our method with EditGuard, the best comparative method in the clean case. We selected three types of video degradation, including Gaussian noise, H.264 video coding, and Poisson noise. As reported in Tab. 4, we found that our $V^2$A-Mark has only slight performance degradation under various degradations compared to the clean scene, and both surpass EditGuard in localization accuracy and copyright reconstruction. Specifically, since we use a multi-frame input, single-frame output structure, which better explores temporal information, our method performs better in handling inter-frame degradation (such as H.264 video coding) than EditGuard which adds watermarks frame by frame. As reported in Tab. 4, the recovered bit accuracy of our method far surpasses EditGuard by 4.18% and 6.95% in QP=5 and QP=10. Meanwhile, our $V^2$A-Mark also outperforms EditGuard by 0.028 and 0.022 in localization accuracy (IoU), which proves its superiority in decoding robustness.

## 4.6 Ablation Studies

To evaluate the contribution of each component, we mainly conduct ablation studies on the temporal alignment and fusion module (TAFM) and degradation prompt learning (DPL). Our results are reported on Tab. 5, where "random degradation" denotes that we randomly select one degradation from Gaussian noise, H.264, and Poisson noise. Comparing case (a) and ours in the "clean" scene, it demonstrates that the proposed TAFM can enhance the localization accuracy and achieve 0.012 gains in IoU, which proves that the proposed TAFM can boost the temporal interaction and realize effective temporal alignment. Comparing case (b) and ours in the "random degradation" scene, due to the learned degradation representations, we find that our method achieves significant gains on localization accuracy and copyright precision.

## 4.7 Applications

Our $V^2$A-Mark can provide focused protection for videos based on user-defined areas. This allows our $V^2$A-Mark to apply to some

| Case | Degradation $\mathcal{D}_v(\cdot)$ | TAFM | DPL | F1 | AUC | IoU | BA(%) |
|---|---|---|---|---|---|---|---|
| (a) | Clean | ✘ | ✔ | 0.935 | 0.962 | 0.885 | 99.47 |
| (b) | Random Degradations | ✔ | ✘ | 0.901 | 0.961 | 0.832 | 98.45 |
| Ours | Clean | ✔ | ✔ | 0.944 | 0.990 | 0.897 | 99.73 |
| | Random Degradations | ✔ | ✔ | 0.912 | 0.975 | 0.849 | 99.43 |

Table 5: Abalation studies on the core parts of $V^2$A-Mark.

global tampering such as visual-audio deepfake. Specifically, we use EfficientSAM [49] to segment the facial regions that need focused protection, and add localization watermarks only to these parts, while still embedding a global copyright watermark. As shown in Fig. 8, we manipulate the identity in the container video frames via SimSwap [5], and alter the first 0.5s of this audio from "there are many jobs for American" to "there are few jobs for American." Subsequently, our $V^2$A-Mark is capable of effectively detecting tampered areas in the face region as well as alterations in the audio. For the audio portion, we determine whether each sample point has been tampered with by evaluating the probability of alteration.

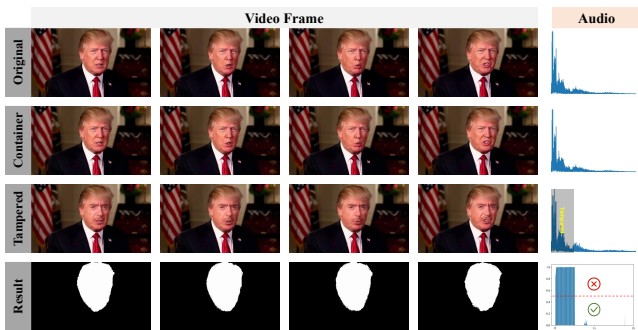

Figure 8: Application scene of the proposed $V^2$A-Mark on the Deepfake tampering [5]. Our $V^2$A-Mark can accurately predict visual tampered masks and the tampered probability of audio samples.

## 5 CONCLUSION

To address the challenges of AI-generated visual-audio forensics, an innovative deep watermarking method with strong generalizability, versatile function, and cross-modal properties dubbed $V^2$A-Mark is proposed. It embeds invisible visual-audio localization and copyright watermarks into the original video frames and audio. If encountering malicious tampering on visual or audio information, we can get accurate tampered visual masks, video copyright, and tampered audio periods in the decoding end via our $V^2$A-Mark. Facing the imminent explosive growth of the AIGC video industry, our $V^2$A-Mark has the potential to safeguard the sustainable development of the AIGC industry, and also establish a clean and transparent information environment.

**Limitations:** Since there is a certain contradiction between the fidelity and robustness of video watermarking, we are still committed to designing advanced modules to achieve better tradeoff. Additionally, as there are few video diffusion-based editing methods available, we have not conducted experiments on larger video editing models. However, we believe our method is robust and effective against all forms of local visual-audio manipulation.

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
