# OpenReview forum: "V2A-Mark: Versatile Deep Visual-Audio Watermarking for Manipulation Localization and Copyright Protection"
_acmmm.org/ACMMM/2024/Conference — MM2024 Poster_

### Official Review · Reviewer_NPry · 2024-05-16

**Rating:** 4
**Confidence:** 3

**Summary:**

This paper proposes V2A-Mark, an innovative watermarking framework for visual-audio manipulation localization and copyright protection. The framework can embed invisible visual-audio localization watermarks and copyright watermarks into the original video frames and audio to allow manipulation localization and copyright protection in the decoding end. Experimental results verify the effectiveness of V2A-Mark as a deep versatile and cross-modal application.

**Strengths:**

(1) For video localization, the framework adopts a temporal alignment and fusion module (TAFM), enabling the temporal correlations of the hidden features.
(2) In the audio section, we embed sample-level versatile watermarks into the pristine audio to identify the tampered samples and extract the copyright information. Furthermore, a cross-modal extraction mechanism is proposed to obtain the final copyright from the information of audio and video frames.
(3) Experimental results on the visual-audio tampering dataset verify the notable merits in localization accuracy generalization abilities, and copyright precision compared to other approaches.

**Limitations:**

(1) In the robustness experiments for watermarking, the QP for video compression is too small to demonstrate the robustness of this method.
(2) The manipulation localization of audio samples is not clearly explained.

**Suitability:**

3

---

### Official Review · Reviewer_Bw2J · 2024-05-18

**Rating:** 4
**Confidence:** 4

**Summary:**

This paper proposes a new multi-modal watermarking method of manipulation localization and copyright protection for video tampering forensics. The method combines the fragility of video-into-video steganography with deep robust watermarking and embed invisible visual-audio localization watermarks and copyright watermarks into the original video frames and audio. Meanwhile, due to the development of TAFM and DPL, the method fully leverages the temporal information of the video and keeps robustness. The effectiveness of the method has been well demonstrated through a series of experiments.

**Strengths:**

This paper innovatively integrates video-into-video steganography with watermarking, utilizing the vulnerability of video-into-video steganography for tampering detection and the robustness of video watermarking for copyright protection, achieving a synergistic effect where the whole is greater than the sum of its parts by accomplishing tampering detection and copyright protection simultaneously. Additionally, in the strategy of watermarks embedding, the proposed TAFM module utilizes the unique temporal correlation of videos to embed information into the temporal domain, which is different from EditGuard and exhibits superior experimental performance. Furthermore, this paper creatively introduces audio watermarking and combines it with video watermarking to achieve cross-modal forensics, offering new insights for this field.

**Limitations:**

To further improve the quality of this paper, I advise the authors to incorporate the following comments.
1.	Using the structure of multi frame input and single frame output, the localization information is embedded into the video via a sliding window, and then a copyright watermark is embedded. Employing this strategy, the same watermark is embedded in all frames. In terms of efficiency, this does not differ from embedding in every single frame, does this lead to low efficiency?
2.	The method proposed indeed demonstrates highly effective detection of tampered regions. However, the consideration of temporal information in this method involves only the preceding and subsequent two frames of the current frame. I am unsure whether this strategy utilizing a sliding window perfectly exploits the unique temporal correlation inherent in videos. Additionally, as proposed in Section 4.2, can it be theoretically proven that the method, due to the utilization of modules such as TAFM, outperforms EditGuard in specific scenarios?
3.	Unlike images, videos are susceptible to frame switching and frame deletion attacks. However, this method does not provide tampering detection results at the frame level. Additionally, experimental results for other digital noises, such as cropping, have not been presented. Should these factors be considered?

**Suitability:**

2

---

### Official Review · Reviewer_cLfE · 2024-05-18

**Rating:** 3
**Confidence:** 4

**Summary:**

This paper proposes the V2A-Mark to address the limitations of current video tampering forensics, such as poor generalizability, singular function, and single modality focus.  Combining the fragility of video-into-video steganography with deep robust watermarking, the V2A-Mark can embed invisible visual-audio localization watermarks and copyright watermarks into the original video frames and audio, enabling precise manipulation localization and copyright protection. However, technological innovation is limited.

**Strengths:**

（1）design an innovative deep versatile, cross-modal video watermarking framework for visual-audio manipulation localization and copyright protection.
（2）develop a temporal alignment and fusion module (TAFM) and a degradation prompt learning (DPL) mechanism, enabling the network to fully leverage temporal information for high-fidelity concealment and robust prediction of localization and copyright results.
（3）Numerous experimental results have verified its effectiveness in localization accuracy, generalization abilities, and copyright precision without any labeled data or additional training required for specific tampering types.

**Limitations:**

(1)  The so-called cross-modal video watermarking framework is essentially a simple combination of video watermarking and audio watermarking, and the relationship between the different modalities is not exploited.

(2) The design of the proposed temporal alignment and fusion module is based on the existing bidirectional cross-attention mechanisms, and moreover, no relevant citation is given.

(3) The audio watermarking section is not described in detail and, according to the paper, it used an existing model directly. Thus, the third contribution is not solid.

(4) There are many parts of the paper that are not clear: What are the invertible blocks? What is an MLP extractor?  Why replicate I\_rec three times? Why and how do you select only the intermediate frames as the final output? As for the use of the term “reference frame”, it will conflict with the one in video coding.

(5) The data comparisons in Table 2 are not very informative because the simulated attacks adopted in their training process are different.

**Suitability:**

3

---

### Official Review · Reviewer_1zSv · 2024-05-24

**Rating:** 4
**Confidence:** 3

**Summary:**

This paper designs an innovative deep versatile, cross-modal video watermarking framework (V2A-Mark) for visual-audio manipulation localization and copyright protection. V2A-Mark combines the fragility of video-into-video steganography with deep robust watermarking to expose the visual tampered area, audio tampered period, and copyright information.
The superiority of V2A-Mark in manipulation localization precision and copyright protection is verified from multiple perspectives.

**Strengths:**

1. A temporal alignment and fusion module (TAFM) and a degradation prompt learning (DPL) mechanism are developed, to enhance the temporal consistency for information concealment and extraction. The degradation prompts of DPL are able to facilitate bit decoding and visual localization.
2. Sample-level versatile watermarks are embedded into the audio to localize the tampered samples and extract the copyright information.
3. The proposed methodology shows better performance in localization and copyright protection abilities. Meanwhile, the security of V2A-Mark outperforms all competitive methods as well.

**Limitations:**

1. It is not explained how robustness is ensured or whether a noise layer is implemented.
2. The method only considers cases where audio modifications do not affect the watermark. The paper does not explain how the copyright information from the audio is used for manipulation localization, and the interaction between modalities is not considered.

**Suitability:**

3

---

### Meta-Review · Area_Chair_vMGT · 2024-07-02

**Recommendation:** Accept (Poster)
**Confidence:** 3

**Metareview:**

The paper proposed an generative AI method for detecting and localizing any visual or audio temperament. Three reviewers suggested "borderline accept" and one reviewer suggested "reject". The primary concern is the novelty of the work. In particular, one reviewer pointed out that the work simply combines watermarking and copyright to protect a video.